# Exploring the Flow Experience and Re-Experience Intention of Students Participating in Water Sports from the Perspective of Regional Tourism and Leisure Environment Suitability



**Zhen Ding** [1]**, Cheng-Ping Li** [2,]*****, Hsiao-Hsien Lin** [3]**, Shen-Te Hung** [4]**, Chih-Hung Tseng** [5] **and Chin-Hsien Hsu** [5,]*****

1  College of Physical Education, Beihua University, Jilin City 132013, China; guhuding@163.com
2  Department of Sports Management, Minghsin University of Science and Technology, Hsinchu 30401, Taiwan
3  School of Physical Education, Jiaying University, Meizhou 514015, China; chrishome12001@yahoo.com.tw
4  Department of Recreation and Sports Management, Tajen University, Pingtung 90741, Taiwan; e10349@tajen.edu.tw
5  Department of Healthcare Industry Technology Development and Management, National Chin-Yi University of Technology, Taichung 41170, Taiwan; boy217010@hotmail.com
*  Correspondence: sm2015@must.edu.tw (C.-P.L.); hsu6292000@yahoo.com.tw (C.-H.H.)

**Abstract:** Previous research on rowing has mostly focused on sports physiology and sports psychology, while the preparation of the competition environment and the participatory behavior of rowers have been less frequently discussed. Therefore, this study intended to discuss the flow experience and revisit the intention of students participating in water-based sports from the perspective of recreational environment fit. Taking the students participating in the Sun Moon Lake Rowing Championships as research subjects, this study conducted a questionnaire survey. A total of 380 questionnaires were sent out and 350 were collected, with a return rate of 92.1%. After excluding 38 invalid questionnaires, 312 valid questionnaires were obtained, and the effective recovery rate was 89.1%. Based on the SPSS and AMOS statistical analyses, the following results were obtained: (1) The needs–supplies fit had a significant impact on flow experience. The path value was 0.60, with a *p*-value of <0.05; (2) the needs–supplies fit had a significant impact on revisit intention. The path value was 0.38, with a *p*-value of <0.05; (3) flow experience had no significant impact on revisit intention. The path value was 0.40, with a *p*-value of >0.05; (4) flow experience had a significant impact on the sense of happiness. The path value was 0.93, with a *p*-value of <0.05; (5) the demands–abilities fit had a significant impact on flow experience. The path value was 0.56, with a *p*-value of <0.05; (6) the demands–abilities fit had a significant impact on revisit intention. The path value was 0.29, with a *p*-value of <0.05; and (7) sense of happiness had no significant impact on revisit intention. The path value was −0.01, with a *p*-value of >0.05. It is suggested that future related studies could focus on the total amount of recreational activities in the water area of Sun Moon Lake. Finally, relevant practical suggestions were made according to the results of this study.

**Keywords:** needs–supplies fit; demands–abilities fit; water-based sports; flow experience; revisit intention

## 1. Introduction

Since the introduction of rowing competitions in Athens at the 1896 Olympic Games, water sports, which originated at Cambridge University in England in 1829, have become widely known and participated in by the public. As a maritime country, Taiwan's geographic location has created many water activity areas. In addition to providing people with water-based recreational activity resources, these water activity areas can also be a niche when developing the tourism industry. On the other hand, with people's increasing demand for sports tourism, outdoor recreational activities are also diversifying, and the development of rowing has become a topic of social concern. As a whole-body movement,

rowing involves the movements of the thighs, the core muscle group, and the upper arm muscle group. Previous research [1] mentioned that rowing is a movement mode that is beneficial to the core and thigh muscle groups. Regarding rowing, Taiwan has improved its competitive strength and competition experience by selecting proper training bases, planning a competition system, cultivating excellent and basic-level rowers, and participating in relevant events.

For a long time, the diversified recreational facilities of Sun Moon Lake have met tourists' demands, both domestically and internationally, and Sun Moon Lake has held many swimming and rowing events. Previous authors [2] pointed out that Sun Moon Lake is a famous tourist attraction, and that Crescent Bay in Sun Moon Lake is an important training base for western rowing and canoeing. With an elevation of 750 m, and benefiting from the altitude and waters, the annual average local temperature is approximately four to five degrees lower than that at sea level. Moreover, the broad water area can be used as an international-standard 2000 m rowing race waterway. Together with the advantages of high-quality water as well as the lack of a tide, current, or strong wind, the training and competition environment can be rated world-class. In addition to sightseeing functions, Sun Moon Lake can also be used for sports tourism. Information on its soft and hard infrastructure and news regarding the site have gained more attention from participants of sports tourism, such as the drowning of rowing players [3], whether water quality samples before, during, and after events conform to drinking water quality and water discharge standards [4], residents' attitudes toward the Cross-lake Swimming Carnivals, and the study on sports tourism impacts and activity support [5]. In other words, in addition to sports tourists' participation in events according to their abilities, whether the soft and hard infrastructure provided by the organizers meets the expectations of the participants is also worthy of exploration. Another study [6] pointed out that the tourist-recreational environment fit refers to the interactive compatibility between the tourists and the recreational environment, and that when the tourists and the environment have similar characteristics, the interaction of the two will produce a good fit.

On the other hand, when participating in challenging events that meet their abilities in the proper competition environment, the participants of sports tourism can devote themselves to the events wholeheartedly, filter out unrelated distractions, and focus on the events, thus generating a flow experience. This psychologically positive feeling of the participants' happiness may influence the evaluation of the activity by the sports tourism participants. Hence, how this psychological feeling affects the participants' willingness to revisit this place to engage in sports tourism has become an issue that the organizers need to pay attention to.

As learned from the literature in the past, most studies on sport-related flow experience function to explore the impact of flow experience on adolescents' intentions to participate in sport [7,8] or to identify what is known about flow states that occur during ad-venture recreation [9], and more often to investigate the relationship between flow states and performance [10], with fewer studies analyzing the ability of leisure sport providers and participants to participate in the sport. On the other hand, most studies focusing on sport-related revisit intentions have focused on the association between team identification and revisit intention [11,12]. Dongfeng [13] explored the association between destination image and the intention to revisit a large-scale tournament from the perspective of foreign tourists. In addition, there are also many related studies exploring the revisit intention of sports participants and spectators from the perspective of consumer behavior [14–16]. However, fewer studies have examined the supply and demand side of leisure sports events or activities as a factor influencing the revisit intention. Most studies on rowing focus on sports physiology and sports psychology [17–20] as well as the impact of rowing equipment on competitions [21–23], but there are few studies analyzing the leadership style of rowing coaches [24,25]. There are also a few explorations into the correlation between the organizers' preparation for the sports competition environment and the participants' participation in local sightseeing activities after the events. For long-term holding and

establishing word-of-mouth, sports tourism activities require an investment in human resources and soft and hard infrastructure, which are important sections the organizers should be concerned about. Therefore, this study treated the students participating in the Sun Moon Lake Rowing Championships as the subjects and discussed their flow experience and revisit intention from the perspective of recreational environment fit. First of all, this paper conducted a literature review on flow experience, environment fit, happiness, and revisit intention, and organized the relevant studies as the theoretical basis of this paper. Taking the participants of the 2022 Sun Moon Lake Rowing Championships as the research subjects, this paper adopted purposive sampling, analyzed the data with AMOS 20.0 statistical software, and then put forward the relevant recommendations according to the research results.

## 2. Literature Review

### 2.1. Flow Experience

In the process of participating in activities, people often strive to achieve a balance between their skills and the difficulty of the activities. Through this balance, they can achieve a leisure experience and the intended leisure benefits. Reference [26] pointed out that a flow experience occurs when people are fully engaged in the work situation and filter out all irrelevant perceptions. In other words, they enter a flow state, which is a temporary and subjective experience, and it is why people are willing to continue to engage in certain activities. Reference [27] further pointed out that a flow experience has two main characteristics—the total concentration in an activity and the enjoyment from the activity, and the effect brought by the flow experience will make a person pay attention to the process rather than the results. This view was later applied to the field of leisure sports to describe similar psychological feelings. In the views of [28], when athletes have a flow experience, they will feel they are strong and unafraid. They can concentrate, integrate their body and mind, and complete their movements without difficulties. Reference [29] further pointed out that a flow experience is a kind of situation. When engaging in an activity with challenges that match their abilities, if people can devote themselves to the activity, concentrate and focus on the activity wholeheartedly, filter out all things irrelevant to the activities, and complete the activities, such an experience can be generated. In general, the psychological feelings generated by people's participation in leisure sports cannot easily be clearly described. Reference [30] further summarized nine relevant characteristics of the flow experience: (1) A challenging activity that requires skills; (2) clear goals; (3) clear feedback; (4) the merging of action and awareness; (5) concentration on the task at hand; (6) a sense of control; (7) loss of self-consciousness; (8) time perception; and (9) experience with purpose. Further research [31] suggested that these nine characteristics can be further divided into three stages. The first stage is the condition for generating the flow, including a clear goal, clear feedback, and proper challenges. The second stage is the current feelings of the flow experience, such as concentration, a sense of control, and a merging of action and awareness. The third stage is the effects brought by the flow experience, including loss of self-consciousness, time perception, and voluntariness.

### 2.2. Environment Fit

Different leisure sports require different environments and spaces. Participants want to pursue mutual coordination and adaptation between individuals and the environment to obtain an impressive experience. For many years, the concept of "fit" has been used in the workplace environment for discussion. In these discussions, the content focuses on an organization and an individual to understand whether the provision of one party can meet the needs of the other party and to further understand the compatibility between the organization and the individual. Reference [32] first applied the concept of fit to the study of outdoor recreational environments. They pointed out that the definition of environment fit refers to the compatibility between the two when at least one party provides what the other party needs. Tourists and environment managers share similar values. From this theory,

when people engage in leisure recreational activities, an individual's ability and the current environmental conditions should integrate closely to create a good fit. Reference [33] further divided recreational environment fit into three categories: Supplementary fit refers to the compatibility achieved when tourists and the managers of recreational areas share similar values; demands–abilities fit refers to the recreational environment demanding tourists to have the abilities of knowledge, skills, and experience; and needs–supplies fit can be discussed from the two dimensions of the interaction between tourists and the activities, the tourists' demand for a recreational environment to include activity attributes, recreational benefits, and recreational experience. From the other perspective of the interaction between tourists and facilities, tourists' demands include natural resources, artificial facilities, and recreational benefits. In this study, demands–abilities fit referred to whether the rowing skills and the strength of Sun Moon Lake Rowing Championship participants could meet the requirements of the events. Moreover, needs–supplies fit referred to whether the competition environment of Sun Moon Lake was appropriate and whether the event organizers' preparation of the moving line, race schedule, or supplies could meet the needs of the participants. As for the measurement of recreational environment fit, most studies adopt the classification of [32], which includes dimensions such as environmental resources, social opportunities, environmental function, environmental facilities, activity knowledge and skills, and business management.

### 2.3. Happiness

The evaluation of an individual's overall life is often constantly revised and adjusted to produce better life satisfaction, and then further improve their sense of happiness. Previous research [34] pointed out that a sense of happiness is to experience the best experience and psychological functions. A sense of happiness is also an element of a good life, which contains two important connotations related to happiness and meaning. Therefore, a sense of happiness is regarded as the positive psychological state of an individual, which is a good feeling produced when a balance is reached internally and externally. Reference [35] evaluated individuals' sense of happiness from six different psychological dimensions, including self-acceptance, positive relations with others, environmental mastery, autonomy, purpose in life, and personal growth. Reference [36] summarized the relevant studies on the sense of happiness and collated four stages that shape the sense of happiness. In the first stage, scholars mostly measure the degree of happiness by the objective external conditions of individuals, such as economic income and family status. In the second stage, the evaluation of the sense of happiness is based on the positive and negative emotions of the individual. Most scholars suggest that when positive emotions are more than negative emotions, happiness will naturally be felt. In the third stage, the factors of individual experience and cognition are gradually valued, so scholars evaluate individuals' feelings on the whole areas of life in the past period. In the fourth stage, in addition to the long-term perception, temporary emotional reactions are also included as a measurement standard for the sense of happiness. Previous research results verify that relevant studies have divided the sense of happiness into a subjective sense of happiness and an objective sense of happiness. Reference [37] stated that studies on the subjective sense of happiness examine the overall evaluation of people's feelings and quality of life, including the cognition and evaluation of positive emotions, negative emotions, and life satisfaction. As for an objective sense of happiness, more emphasis is given to people's reactions and cognition of their living conditions.

### 2.4. Revisit Intention

After using leisure recreational environment facilities and activities, people will often have a certain psychological evaluation, which will further affect whether they will participate in activities again in the future. Reference [38] pointed out that the revisit intention of tourists refers to whether they are willing to go to certain recreation areas again. Reference [39] further suggested that a satisfactory recreational experience is not only the

presentation of the tourists' internal feelings on their recreation participation but also an effective predictor of the revisit intention. Therefore, the first way to improve tourists' revisit intention is to improve their recreational experience. Reference [40] pointed out that revisit intention is an important indicator to measure the images of tourist attractions and customer satisfaction. In other words, if sports tourism activities can provide participants with a complete participation experience that improves their satisfaction, they may choose to participate in the activity again in the future. Reference [41] indicated that revisit intention often appears in the relevant literature on recreation satisfaction, for example, in the discussion on recreational areas, indicating the revisit intention or the repurchase intention of tourism products. When tourists choose to enjoy a recreational area, the recreational opportunities, environmental facilities, and activity and service evaluations must meet their requirements, which may also increase their revisit intention. The derivative behaviors of revisit intention may include customer recommendations, public recommendations, and word of mouth.

With the gradual slowing down of the COVID-19 pandemic, studies on revisit intention of leisure sports in the post-pandemic era have taken on a different appearance, with more studies incorporating the COVID-19 pandemic theme. For example, [42] explored sports fans' nostalgia and revisit intentions to stadiums during the COVID-19 pandemic. The research results indicated that nostalgia for sports teams significantly influenced sports fans' intention to revisit stadiums. Nostalgia for sports teams encouraged fans who attended live sports matches during the pandemic more than fans who did not attend the live matches. Boo and Kim [43] examined the likelihood of return for customers who have prior event experience and found that trust in events is a strong predictor of event revisit intention, while perceived risk mediates the relationship between trust and revisit intention, albeit weakly. It can be seen that with a change in social environment, the factors affecting leisure sports participants' revisit intention are more diversified.

*2.5. Relationships among Variables*

As for the relationships among variables, the research conducted by [44] on participants in the Cross Penghu Bay Swim showed the needs–supplies fit has a significant impact on flow experience. Reference [45] analyzed tourists in Xi'an's Cuihuashan National Geopark and further verified that the needs–supplies fit has a significant impact on revisit intention. Reference [46] conducted an investigation on participants of urban adventure games and pointed out that flow experience has a significant impact on revisit intention. Reference [47] studied the flow experience and sense of happiness of middle-aged and elderly people participating in a triathlon, and the results showed that the flow experience has a significant impact on their sense of happiness. Reference [48] studied cyclists' recreation specialization, the tourists–recreation environment fit, and flow experience. Their results showed the demands–abilities fit has a significant impact on the flow experience. Moreover, the demands–abilities fit was found to have a significant impact on revisit intention, according to a previous study [44]. Reference [49] studied tourists taking a trip from the Chinese mainland to Macao and found that a sense of happiness has a significant impact on revisit intention.

## 3. Research Method

*3.1. Research Structure*

This study discussed the flow experience and revisit intention of students participating in water-based sports from the perspective of recreational environment fit. According to the literature discussion, the research structure is shown in Figure 1.

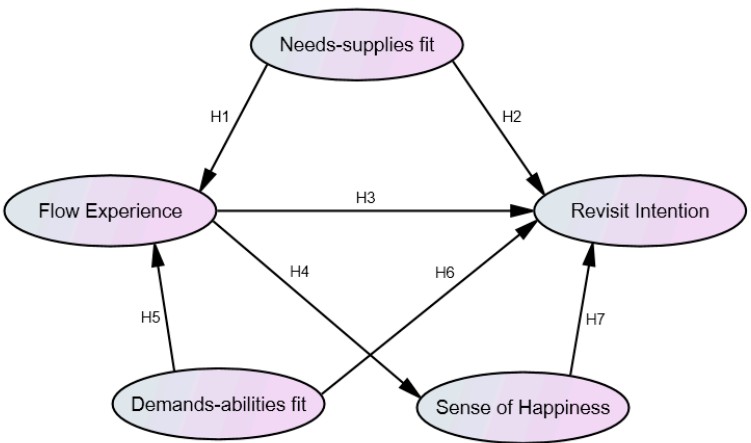

**Figure 1.** Research diagram.

*3.2. Research Hypotheses*

According to the literature review and the research structure, the following hypotheses were proposed:

**Hypothesis H1.** *The needs–supplies fit has a significant impact on flow experience.*

**Hypothesis H2.** *The needs–supplies fit has a significant impact on revisit intention.*

**Hypothesis H3.** *Flow experience has a significant impact on revisit intention.*

**Hypothesis H4.** *Flow experience has a significant impact on a sense of happiness.*

**Hypothesis H5.** *The demands–abilities fit has a significant impact on flow experience.*

**Hypothesis H6.** *The demands–abilities fit has a significant impact on revisit intention.*

**Hypothesis H7.** *Sense of happiness has a significant impact on revisit intention.*

*3.3. Research Subjects*

This study took students participating in the 2022 Sun Moon Lake Rowing Championships as subjects and adopted purposive sampling to gather data. The questionnaires were released in the form of a Google Forms questionnaire from 11 to 15 July 2022. A total of 380 questionnaires were released, and 350 responses were obtained. After excluding incomplete or unclear responses, a total of 312 valid questionnaires were obtained, for an effective recovery rate of 89.1%.

*3.4. Research Tools*

The content of the questionnaire was mainly based on the relevant literature of [32,33,44,50–52], with reference to the research questionnaires of [53–56]. The questionnaire content was modified to meet the needs of this study. The questionnaire contained five sections with a total of 48 items, including 7 items for basic personal information, 19 items for recreational environment fit (needs–supplies fit and demands–abilities fit), ten items for a sense of happiness, nine items for flow experience, and three items for revisit intention. Except for the demographic variables, which adopted a nominal scale, all items used a seven-point Likert scale, with answers ranging from "strongly disagree" to "strongly agree" represented by scores of 1 to 7, respectively. Confirmatory factor analysis (CFA) was used in this study to measure the convergent validity of this scale, and the bootstrap confidence interval method was used to examine the dimensional discriminant validity of the questionnaire.

### 3.5. Data Processing and Analysis

This study sorted out and reviewed the questionnaires collected after the completion of the activity. The valid questionnaires were manually coded, and SPSS 22.0 and AMOS 20.0 statistical software were used as the data analysis tools to review the data content.

## 4. Research Results

### 4.1. Sample Characteristics

According to Table 1, among the 312 respondents, most were males with a senior and vocational high school educational level. Their best rowing achievements mostly ranged from first to fourth place and occurred at national events, and their rowing experience was mostly under four years. They mostly lived in central Taiwan, mostly had no experience in international events, and most had used stroke rate monitors nine or more times.

**Table 1.** Characteristics of the respondents.

| Background Variable | Classification Standard | Number of Samples | Percentage % | Cumulative Percentage % |
|---|---|---|---|---|
| Gender | Male | 247 | 79.2 | 79.2 |
| | Female | 65 | 20.8 | 100.0 |
| Current level | Junior high school | 45 | 14.4 | 14.4 |
| | Senior and vocational high school | 116 | 37.2 | 51.6 |
| | University and college | 143 | 45.8 | 97.4 |
| | Graduate institute | 8 | 2.6 | 100.0 |
| Best rowing achievements | National events ranking 1–4 | 163 | 52.2 | 52.2 |
| | National events ranking 5–8 | 64 | 20.5 | 72.8 |
| | Have not won | 85 | 27.2 | 100.0 |
| Years of rowing | Under one year | 84 | 26.9 | 26.9 |
| | Under four years | 123 | 39.4 | 66.3 |
| | Under seven years | 67 | 21.5 | 87.8 |
| | Over eight years | 38 | 12.2 | 100.0 |
| Residential place | Northern Taiwan | 83 | 26.6 | 26.6 |
| | Central Taiwan | 136 | 43.6 | 70.2 |
| | Southern Taiwan | 40 | 12.8 | 83.0 |
| | Eastern Taiwan | 31 | 9.9 | 92.9 |
| | Offshore islands | 22 | 7.1 | 100.0 |
| Experience in international events | Yes | 93 | 29.8 | 29.8 |
| | No | 219 | 70.2 | 100.0 |

### 4.2. Measurement Mode Analysis

The reliability and validity of the questionnaire were verified and analyzed by confirmatory factor analysis (CFA). Reference [57] pointed out that a high modification indices (MI) value indicates the measurement errors between subjects are correlated and will affect the scale. Therefore, items with high MI values were deleted, including Item 19 in recreational environment fit, Items 1 and 7 in sense of happiness, Item 4 in flow experience, Item 3 in social opportunities, Items 1 and 5 in confidence, and Item 1 in promotion.

(1)   Verification of convergence validity

CFA was used to measure the convergence validity of the scales in this study. The t-value was significant, the standard factor load was between 0.55 and 0.88, and the

CR value was between 0.74 and 0.89. These values conformed to the values suggested by [58–60], who stated the standard factor load must exceed 0.50 and the CR must exceed 0.60. The AVE of the present study was between 0.51 and 0.72, which conformed to the standard of [59], who suggested the AVE should be over 0.50. The above findings indicated the research scale conformed with the convergence validity, as shown in Tables 2–6.

**Table 2.** Summary of confirmatory factor analysis—needs–supplies fit.

| Potential Variable | Manifest Variable | Non-Standard Factor Load | Standard Error | C.R | *p* | Factor Load | SMC | Composite Reliability | AVE |
|---|---|---|---|---|---|---|---|---|---|
| **Model Parameter Estimation** | | | | | | **Convergent Validity** | | | |
| Environmental resources | Fit 1 | 1.00 | | | | 0.81 | 0.66 | 0.85 | 0.66 |
| | Fit 2 | 1.00 | 0.06 | 16.17 | *** | 0.84 | 0.70 | | |
| | Fit 3 | 1.04 | 0.07 | 15.16 | *** | 0.80 | 0.64 | | |
| Social opportunities | Fit 4 | 1.00 | | | | 0.78 | 0.62 | 0.82 | 0.60 |
| | Fit 5 | 1.03 | 0.07 | 14.27 | *** | 0.79 | 0.63 | | |
| | Fit 6 | 0.99 | 0.07 | 13.52 | *** | 0.76 | 0.58 | | |
| Environmental function | Fit 7 | 1.00 | | | | 0.82 | 0.68 | 0.86 | 0.67 |
| | Fit 8 | 0.85 | 0.05 | 15.49 | *** | 0.81 | 0.65 | | |
| | Fit 9 | 0.95 | 0.06 | 16.36 | *** | 0.83 | 0.69 | | |
| Environmental equipment | Fit 10 | 1.00 | | | | 0.60 | 0.35 | 0.75 | 0.51 |
| | Fit 10 | 1.04 | 0.11 | 9.85 | *** | 0.75 | 0.56 | | |
| | Fit 12 | 1.03 | 0.10 | 10.19 | *** | 0.78 | 0.61 | | |

Source: Compiled by this study. *** *p* < 0.001.

**Table 3.** Summary of confirmatory factor analysis—demands–abilities fit.

| Potential Variable | Manifest Variable | Non-Standard Factor Load | Standard Error | C.R | *p* | Factor Load | SMC | Composite Reliability | AVE |
|---|---|---|---|---|---|---|---|---|---|
| **Model Parameter Estimation** | | | | | | **Convergent Validity** | | | |
| Activity knowledge and skills | Fit 13 | 1.00 | | | | 0.55 | 0.30 | 0.82 | 0.54 |
| | Fit 14 | 1.23 | 0.13 | 9.30 | *** | 0.77 | 0.60 | | |
| | Fit 15 | 1.22 | 0.13 | 9.48 | *** | 0.79 | 0.63 | | |
| | Fit 16 | 1.22 | 0.13 | 9.54 | *** | 0.81 | 0.66 | | |
| Business management | Fit 17 | 1.00 | | | | 0.73 | 0.53 | 0.74 | 0.59 |
| | Fit 18 | 1.03 | 0.08 | 12.34 | *** | 0.81 | 0.65 | | |

Source: Compiled by this study. *** *p* < 0.001.

**Table 4.** Summary of confirmatory factor analysis—flow experience.

| Potential Variable | Manifest Variable | Non-Standard Factor Load | Standard Error | C.R | *p* | Factor Load | SMC | Composite Reliability | AVE |
|---|---|---|---|---|---|---|---|---|---|
| **Model Parameter Estimation** | | | | | | **Convergent Validity** | | | |
| Sense of control | Experience 1 | 1.00 | | | | 0.81 | 0.66 | 0.86 | 0.67 |
| | Experience 2 | 1.00 | 0.06 | 16.09 | *** | 0.81 | 0.66 | | |
| | Experience 3 | 0.99 | 0.06 | 16.47 | *** | 0.85 | 0.72 | | |
| Concentration | Experience 5 | 1.00 | | | | 0.82 | 0.67 | 0.83 | 0.72 |
| | Experience 6 | 1.13 | 0.07 | 15.10 | *** | 0.88 | 0.77 | | |
| Time perception | Experience 7 | 1.00 | | | | 0.82 | 0.67 | 0.85 | 0.65 |
| | Experience 8 | 0.96 | 0.06 | 15.67 | *** | 0.80 | 0.64 | | |
| | Experience 9 | 0.94 | 0.06 | 15.69 | *** | 0.81 | 0.66 | | |

Source: Compiled by this study. *** *p* < 0.001.

**Table 5.** Summary of confirmatory factor analysis—sense of happiness.

| | | Model Parameter Estimation | | | | Convergent Validity | | | |
|---|---|---|---|---|---|---|---|---|---|
| **Potential Variable** | **Manifest Variable** | **Non-Standard Factor Load** | **Standard Error** | **C.R** | ***p*** | **Factor Load** | **SMC** | **Composite Reliability** | **AVE** |
| Happy mood | Sense of happiness 2 | 1.00 | | | | 0.86 | 0.74 | 0.89 | 0.67 |
| | Sense of happiness 3 | 0.98 | 0.05 | 18.21 | *** | 0.82 | 0.68 | | |
| | Sense of happiness 4 | 0.91 | 0.05 | 18.66 | *** | 0.84 | 0.70 | | |
| | Sense of happiness 5 | 0.85 | 0.05 | 16.05 | *** | 0.76 | 0.58 | | |
| Life satisfaction | Sense of happiness 6 | 1.00 | | | | 0.82 | 0.67 | 0.87 | 0.63 |
| | Sense of happiness 8 | 0.89 | 0.07 | 13.68 | *** | 0.71 | 0.51 | | |
| | Sense of happiness 9 | 0.92 | 0.05 | 16.80 | *** | 0.83 | 0.70 | | |
| | Sense of happiness 10 | 0.96 | 0.06 | 16.27 | *** | 0.81 | 0.65 | | |

Source: Compiled by this study. *** $p < 0.001$.

**Table 6.** Summary of confirmatory factor analysis—revisit intention.

| | | Model Parameter Estimation | | | | Convergent Validity | | | |
|---|---|---|---|---|---|---|---|---|---|
| **Potential Variable** | **Manifest Variable** | **Non-Standard Factor Load** | **Standard Error** | **C.R** | ***p*** | **Factor Load** | **SMC** | **Composite Reliability** | **AVE** |
| Revisit intention | Intention 1 | 1.00 | | | | 0.84 | 0.71 | 0.85 | 0.66 |
| | Intention 2 | 0.91 | 0.06 | 14.89 | *** | 0.86 | 0.74 | | |
| | Intention 3 | 0.80 | 0.06 | 13.53 | *** | 0.74 | 0.54 | | |

Source: Compiled by this study. *** $p < 0.001$.

(2)  Discriminant validity

This study used the bootstrap confidence interval method to verify the dimensional discriminant validity. The estimation was repeated 1000 times at a 95% confidence interval to calculate the correlation coefficient between different dimensions. If the value of 1 was not included, the dimensions would have discriminant validity [61], as shown in Tables 7–10.

**Table 7.** Bootstrap correlation coefficient 95% confidence interval—needs–supplies fit.

| | | | Bias-Corrected | | | Percentile Method | |
|---|---|---|---|---|---|---|---|
| | | | **Estimation** | **Lower Limit** | **Upper Limit** | **Lower Limit** | **Upper Limit** |
| Environmental resources | <--> | Social opportunities | 0.91 | 0.85 | 0.97 | 0.85 | 0.97 |
| Environmental resources | <--> | Environmental function | 0.72 | 0.60 | 0.81 | 0.60 | 0.81 |
| Environmental resources | <--> | Environmental equipment | 0.84 | 0.74 | 0.92 | 0.74 | 0.92 |
| Social opportunities | <--> | Environmental function | 0.69 | 0.57 | 0.80 | 0.57 | 0.80 |
| Social opportunities | <--> | Environmental equipment | 0.88 | 0.79 | 0.97 | 0.79 | 0.97 |
| Environmental function | <--> | Environmental equipment | 0.85 | 0.76 | 0.92 | 0.76 | 0.93 |

Source: Compiled by this study.

**Table 8.** Bootstrap correlation coefficient 95% confidence interval—demands–abilities fit.

| | | | Bias-Corrected | | | Percentile Method | |
|---|---|---|---|---|---|---|---|
| | | | Estimation | Lower Limit | Upper Limit | Lower Limit | Upper Limit |
| Knowledge and skills of activities | <--> | Business management | 0.90 | 0.80 | 0.99 | 0.80 | 0.99 |

Source: Compiled by this study.

**Table 9.** Bootstrap correlation coefficient 95% confidence interval—flow experience.

| | | | Bias-Corrected | | | Percentile Method | |
|---|---|---|---|---|---|---|---|
| | | | Estimation | Lower Limit | | Estimation | Lower Limit |
| Sense of control | <--> | Concentration | 0.79 | 0.68 | 0.89 | 0.68 | 0.89 |
| Sense of control | <--> | Time perception | 0.88 | 0.79 | 0.95 | 0.79 | 0.95 |
| Concentration | <--> | Time perception | 0.75 | 0.63 | 0.85 | 0.63 | 0.86 |

Source: Compiled by this study.

**Table 10.** Bootstrap correlation coefficient 95% confidence interval—sense of happiness.

| | | | Bias-Corrected | | | Percentile Method | |
|---|---|---|---|---|---|---|---|
| | | | Estimation | Lower Limit | Upper Limit | Lower Limit | Upper Limit |
| Happy mood | <--> | Life satisfaction | 0.94 | 0.90 | 0.98 | 0.90 | 0.98 |

Source: Compiled by this study.

(3)    Structure mode analysis

In terms of the fitness analysis of this research mode, this study referred to the suggestions of [58,62] and conducted a fitness analysis with seven indices. In the mode of the present study, the ratio of $\chi^2$ to the degree of freedom after modification was less than 5 (3.00). Reference [58] pointed out that the closer the value of GFI and AGFI is to 1, the better. In the current study, after modification, the values of GFI and AGFI were 0.80 and 0.74, respectively. Reference [63] mentioned that an RMSEA value between 0.05 and 0.08 indicates that the mode is good, and the RMSEA value of the present study was 0.08. The CFI value should be over 0.90, and the CFI value after modification in the current study was 0.90. The PCFI value should be over 0.50, and the PCFI value after modification was 0.80 in the present study, indicating the overall fitness of this study was within the standard scope. Hence, the results of this study were acceptable, as shown in Table 11.

**Table 11.** Fitness analysis of the overall mode of this study.

| Fit Indices | Acceptable Scope | Research Mode of This Study | Mode Fitness Judgment |
|---|---|---|---|
| $\chi^2$ (Chi-square) | The smaller, the better | 1873.12 | |
| Ratio of $\chi^2$ to the degree of freedom | <3.0 | 3.00 | Conformed |
| GFI | >0.80 | 0.80 | Conformed |
| AGFI | >0.80 | 0.74 | Acceptable |
| RMSEA | <0.08 | 0.08 | Conformed |
| CFI | >0.90 | 0.90 | Conformed |
| PCFI | >0.50 | 0.80 | Conformed |

It can be seen from Table 12 and Figure 2 that H1 was supported. In other words, the needs–supplies fit had a significant impact on flow experience. This result was the same as that of [44]. This result could be due to the organizer of the Sun Moon Lake Rowing Championships providing complete services and venues that allowed the student participants to make full play of their strengths, challenge themselves, and obtain the flow experience. H2 was supported. In other words, the needs–supplies fit had a significant impact on revisit intention. This result was the same as that of [45], possibly because the student participants of the Sun Moon Lake Rowing Championships identified with the event venues and planning measures and intended to participate again. H3 was not supported. In other words, flow experience had no significant impact on revisit intention, which matched the results of [64]. Although the student participants may have achieved the flow experience, their intention to participate again could be influenced by variables such as schedule and physical limitations. H4 was supported. In other words, flow experience had a significant impact on the sense of happiness. This result was the same as that of [65]. A possible reason may be that the flow experience resulted in the loss of self-consciousness and concentration, and this positive psychological state brought the student participants good feelings. H5 was supported. In other words, the demands–abilities fit had a significant impact on flow experience. This result was similar to that of [48]. After long-term training, the physical strength and skills of the student participants would have reached a certain level, so they would have a better chance to achieve their desired results in real competitions, thus improving the degree of flow experience. H6 was supported. In other words, the demands–abilities fit had a significant impact on revisit intention. This result was the same as that of [44]. When the student participants could prepare well for the events, they would be more likely to believe they could achieve their goals. Once they knew they could achieve these goals, they would be more likely to participate in future events. H7 was not supported. In other words, a sense of happiness had no significant impact on revisit intention. This result was different from that of [66], possibly because although the student participants felt happy due to the positive psychological feelings obtained from participating in the events, and whether they would participate in future events depended on many factors. Therefore, they may not choose to revisit it in the future.

**Table 12.** Empirical results of research hypotheses.

| Hypotheses | Path Value | *p* Value | Verification Result |
|---|---|---|---|
| H1: The needs-supplies fit has a significant impact on flow experience. | 0.60 | *** | Supported |
| H2: The needs-supplies fit has a significant impact on revisit intention. | 0.38 | *** | Supported |
| H3: Flow experience has a significant impact on revisit intention. | 0.40 | 0.13 | Not supported |
| H4: Flow experience has a significant impact on a sense of happiness. | 0.93 | *** | Supported |
| H5: The demands-abilities fit has a significant impact on flow experience. | 0.56 | *** | Supported |
| H6: The demands-abilities fit has a significant impact on revisit intention. | 0.29 | 0.02 | Supported |
| H7: Sense of happiness has a significant impact on revisit intention. | −0.01 | 0.96 | Not supported |

*** $p < 0.001$.

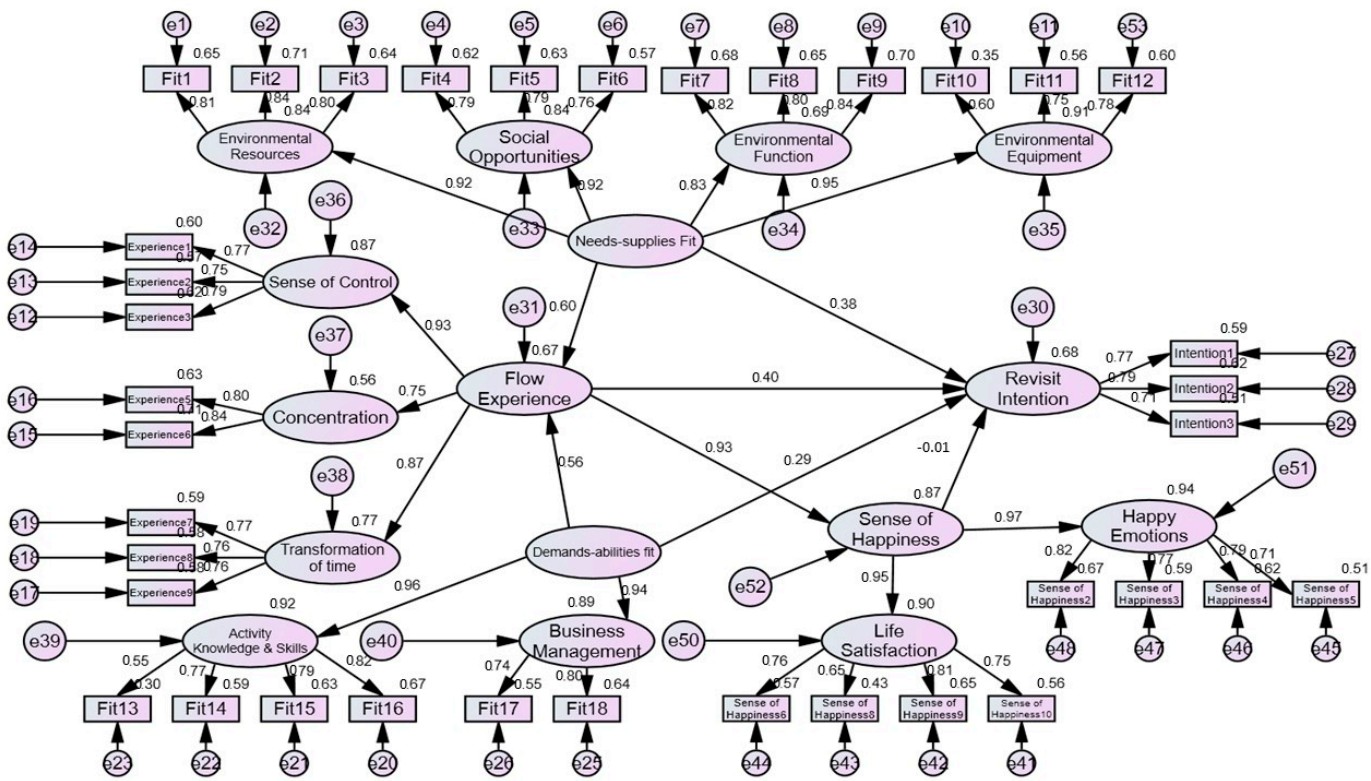

**Figure 2.** Empirical study on the recreational environment fit, flow experience, and revisit intention of students participating in the rowing championship.

## 5. Conclusions

This study found the needs–supplies fit and the demands–abilities fit had a significant impact on the student participants' flow experience and revisit intention, and their flow experience also had a significant impact on their sense of happiness. However, the student participants' flow experience and sense of happiness had no significant impact on their revisit intention. These conclusions revealed the hypotheses of this study to be both rigorous and accurate. In addition, this study complemented the deficiencies of previous studies on revisit intention, which mostly used the variables of attractiveness, place attachment, service quality, brand image, participating motivation, and destination imagery for the research. Especially the fact that leisure sports participants were more concerned about whether they could coordinate and adapt to the leisure sports environment than general tourists, the environment fit combined with the leisure sports facility providers and participants, as well as the ability of the participants and the compatibility of the venues, were incorporated into the study to obtain a more complete analysis.

## 6. Recommendations

(1)    Regarding the student participants

H1 and H2 were both supported; in other words, the demands–abilities fit had a significant impact on flow experience and revisit intention. Hence, it is suggested that the student participants could improve their professional skills and actively study relevant professional knowledge. For example, in terms of physiology, through kinematics, the student participants can understand the catch and finish stages of their stroke and analyze the movement range of their joints. By doing so, the student participants can further understand the angle of ankle extension during the catch and finish stages to improve their output power and rowing speed. Psychology is also an important factor that affects rowing performance. Therefore, it is suggested that student participants also use mental training methods. In particular, imagery training plays an important role in mental training.

By focusing on integrating imagery training into the self-training schedule, and with the addition of technology improvements, physical training, and the output of consistent rowing actions, these measures can not only help enhance performance but also generate flow experience due to the balance of the individual's abilities and challenges. On the other hand, when the student participants' abilities are consistent with their challenges, in addition to facing the challenges of the events and achieving satisfying performance, they may also have more time to relax by sightseeing near the event venue because of their sufficient preparation. Further, they will connect the competition results with the venue, thereby improving the revisit intention of the participants.

H4 was supported; in other words, flow experience had a significant impact on the sense of happiness. It is suggested that if the student participants want to gain a joyful flow experience, they should seek higher challenges and improve their personal skills. When personal training has goals and achievements, it is possible to improve the satisfaction of the overall life condition, further develop personal cognition, emotions, and physical and mental health, and finally obtain a sense of happiness.

(2)    Regarding rowing championship organizers

The results of this study indicated that the needs–supplies fit had a significant impact flow experience and revisit intention. Therefore, it is suggested that rowing championship organizers engage in proper yearly planning before holding events. For example, a complete service system that addresses issues related to transportation, connections, accommodation, and traffic flow can be built. By providing a complete infrastructure, championships can facilitate the student participants in giving play to their strengths, obtaining a sense of enjoyment due to concentration on the events, and creating a flow experience. On the other hand, it is suggested that rowing championship organizers further cooperate with the community to connect the surrounding scenic spots through soft power, such as introducing meals with local flavors to the student participants in combination with the community resources or providing handicrafts with local characteristics as souvenirs. These measures could not only improve the quality of the events but also enable the student participants to value and enjoy the process of the events rather than just the results. Such processes could also help the student participants create a flow experience. As mentioned above, the student participants may also take part in sightseeing activities while participating in events. Therefore, organizers should carefully consider how to create a desire for the student participants to revisit in the future. It is suggested that the natural and cultural resources of Sun Moon Lake be introduced into the rowing championships by presenting marketing information about the local geology, topography, lifestyle, and aesthetics. Organizers can also try to negotiate cooperation with Sun Moon Lake PLUS, an alliance. In particular, this alliance has 5 guilds, 28 associations, 3 large-scale tourism and entertainment attractions (Formosan Aboriginal Culture Village, Atayal Resort, and Sun-Link-Sea Forest and Nature Resort), and 3 passenger transport companies. Rowing championships can be promoted in cooperation with various development or marketing itineraries launched by the alliance, which will help to improve the image of Sun Moon Lake Rowing Championships, further, give the student participants a sense of identity, and make them willing to return to Sun Moon Lake in the future to participate in the competition and take part in sightseeing activities again.

(3)    Regarding research in the future

With the gradual recovery of domestic tourism after the COVID pandemic, the number of participants in outdoor sports is expected to increase. Located in central Taiwan, Sun Moon Lake has become a popular place for water activities. In addition to sightseeing yachts, canoes and stand-up paddles are also attracting more water enthusiasts. With this trend, issues in the management of Sun Moon Lake's water activities are bound to appear again. Therefore, it is suggested that future studies conduct research on the total number of Sun Moon Lake aquatic recreation activities. In particular, the total number of recreation activities will change according to the actual environmental change of Sun Moon Lake,

and individuals' cognition of the total amount of recreation activities will vary due to the recreation quality and service level of Sun Moon Lake's water activities. In other words, it is important to analyze the carrying capacity of Sun Moon Lake's water activities, which influences rowers' training experience and water activity participation, as well as their intention to revisit as tourists. Therefore, if follow-up research can conduct an analysis of this issue, it will make some contributions.

**Author Contributions:** Conceptualization, Z.D.; Methodology, C.-P.L.; Software, C.-P.L.; Formal analysis, C.-P.L., H.-H.L. and C.-H.T.; Investigation, H.-H.L. and S.-T.H.; Resources, S.-T.H.; Data curation, C.-H.T.; Writing—original draft, C.-P.L. and C.-H.H.; Writing—review & editing, C.-P.L. and C.-H.H.; Supervision, Z.D.; Project administration, Z.D.; Funding acquisition, Z.D. All authors have read and agreed to the published version of the manuscript.

**Funding:** This research received no external funding.

**Institutional Review Board Statement:** During the research and investigation process, we pointed out in Announcement No. 1010265075 of the Health Department of the Taiwan Executive Yuan that if the case has fully informed the subjects of the investigation method and purpose, the non-human test measurement method will use the questionnaire survey method. And voluntary manner. In the circumstances indicated above, we consider this manuscript suitable for exemption from ethical review.

**Informed Consent Statement:** Written informed consent has been obtained from the participants in this study.

**Data Availability Statement:** The data will not be made public, and its use and access rights will be managed by the corresponding author.

**Conflicts of Interest:** The authors declare no conflict of interest.

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
