# Peer review of "Exploring the Flow Experience and Re-Experience Intention of Students Participating in Water Sports from the Perspective of Regional Tourism and Leisure Environment Suitability"

_sustainability, doi:10.3390/su151914614_

Round 1
Reviewer 1 Report
Dear Authors,
Your paper exhibits relevance and adherence to scientific writing norms, although it necessitates minor revisions to elevate its overall quality. My observations and recommendations are outlined below:
The title is lengthy; it should be shortened.
The abstract is not well-structured. The authors are invited to clarify the research gap they intend to address, provide a brief description of the methodology used, and outline future research directions.
In the last paragraph of the introduction, the authors should outline the paper's structure.
I recommend adding sub-headings in the literature review section, which will help readers better comprehend previous research.
The research method adopted is relevant and aligns with the paper's objectives, but it's not well-described. The authors are encouraged to justify the techniques and approaches used and explain their suitability within the research context.
The results are well-presented.
The discussion section should include a discussion related to the obtained results with necessary comparisons.
The conclusion should be separated from suggestions and should encompass limitations and the main contributions of your research in advancing knowledge in the field.
"Suggestions" can be replaced by "implications" or "research agenda."
There are too few references for such a hot topic. The authors should add more pertinent and recent papers to enhance the overall quality.
Author Response
Reviewers’ Comments and Suggestions for Author (Round 1)
29 September 2023
Dear Reviewer,
Thank you for the constructive suggestions and comments on our manuscript(ID:Sustainability-2621888). The suggestions and comments are helpful for improving the manuscript. We are submitting the revised version of the manuscript with our responses to the suggestions and comments by the reviewer. Many thanks for your guidance.
Our responses to each suggestion and comment are as follows, and they are presented in blue texts with a grey background color in the revised manuscript.
Response:
Thank you very much for your comments and suggestion. The modifications are as follows:

Reviewer 2 Report
The article investigates the impact of demands-abilities fit and needs-supplies fit on the flow experience and revisit intention of student participants in water-based sports, highlighting the importance of aligning individual skills and challenges to enhance athletes' experiences, with recommendations for both athletes and event organizers. While the study builds upon existing literature, particularly in the areas of flow experience and revisit intention, it introduces the concept of "recreational environment fit" to water-based sports, which demonstrates a certain level of originality in the study's approach. Overall, the article is well-structured and demonstrates a rigorous approach to research. It could benefit from some minor improvements in areas like providing more recent literature references and offering specific recommendations for each stakeholder group. However, these are relatively minor points in an otherwise well-executed study:
Abstract: it could benefit from more specific numerical results to give readers a clearer sense of the study's outcomes.
Introduction: it could provide more background information on the existing literature regarding flow experience and revisit intention in sports to establish the research gap.
Literature Review: The literature review provides a solid foundation for the study by discussing relevant concepts, such as flow experience, revisit intention, and recreational environment fit. It effectively integrates previous research findings into the current study's framework. Yet, there is a lack of recent literature, and the article could benefit from incorporating more up-to-date research to enhance its relevance. I suggest:
Santos, E., & Castanho, R. A. (2022). The impact of size on the performance of transnational corporations operating in the textile industry in Portugal during the COVID-19 pandemic. Sustainability, 14(2), 717.
To discuss the impact of external factors, such as the COVID-19 pandemic, on the performance of organizations, which can be analogous to the potential external factors affecting rowing and outdoor sports.
Santos, E., Lisboa, I., & Eugénio, T. (2022). The financial performance of family versus non-family firms operating in nautical tourism. Sustainability, 14(3), 1693.
To provide insights into the financial performance of firms in the tourism sector, including nautical tourism, which can be valuable when discussing the financial aspects of rowing and outdoor sports.
Method: include more details on how the questionnaires were developed or adapted from existing sources.
Results: providing effect sizes or confidence intervals alongside significance tests would enhance the interpretability of the results.
Discussion: it could provide more detailed comparisons with previous studies to enrich the discussion.
Conclusion and Suggestions: The recommendations provide practical insights for student participants, championship organizers, and future research directions. However, it would be beneficial to include specific, actionable recommendations for each stakeholder group.
Author Response

(The authors gave the same response as above.)
